Management of calcified canals during root canal treatment. A systematic review of case reports

Giri Kunal 1
Banga Kulvinder 1
Arora Suraj 2
Elmsmari Firas 3 4 f.elmsmari@ajman.ac.ae
http://orcid.org/0000-0003-3985-5674 Pawar Ajinkya M. 1 ajinkya@drpawars.com
1 Department of Conservative Dentistry and Endodontics, Nair Hospital Dental College , Mumbai, Maharashtra , India
2 Department of Restorative Dental Sciences, King Khalid University , Abha , Saudi Arabia
3 Department of Clinical Sciences, College of Dentistry, Ajman University of Science & Technology , Al Jerf, Ajman , United Arab Emirates
4 Center of Medical and Bio-Allied Health Sciences Research, Ajman University of Science & Technology , Al Jerf, Ajman , United Arab Emirates
Abu Hasna Amjad
Electronic publication date: 2025 Sep 1
Publication date: 2025
Volume: 13
Electronic Location ID: e19900
Received 2025 Apr 25; Accepted 2025 Jul 22
Copyright: © 2025 Giri et al.
Copyright year: 2025
Copyright holder: Giri et al.
License: This is an open access article distributed under the terms of the Creative Commons Attribution License, which permits unrestricted use, distribution, reproduction and adaptation in any medium and for any purpose provided that it is properly attributed. For attribution, the original author(s), title, publication source (PeerJ) and either DOI or URL of the article must be cited.
License URL: https://creativecommons.org/licenses/by/4.0/

Keywords: Root canal therapy, Calcification, Cone-beam computed tomography, Chelating agents, Non-vital teeth

Funding: Deanship of Research and Graduate Studies at King Khalid University, Abha, Saudi Arabia RGP2/618/46 This work was supported by the Deanship of Research and Graduate Studies at King Khalid University, Abha, Saudi Arabia through the Large Research Group initiative, grant number (RGP2/618/46). The funders had no role in study design, data collection and analysis, decision to publish, or preparation of the manuscript.

==============================
Background and Objectives

Calcified canals pose substantial challenges in endodontic treatment because of narrowing of pulp chambers and canal lumens, reducing accessibility and biomechanical preparation. Nevertheless, with advanced techniques and well-devised strategies, successful management of calcified canals is achievable. This comprehensive analysis aims to rigorously assess extant literature concerning the clinical management of teeth affected by pulp canal obliteration that necessitates root canal therapy and to formulate a comprehensive, updated algorithm for clinical decision-making strategies.

Materials and Methods

The study adhered to the PRISMA 2020 guidelines, the Cochrane Handbook (version 5.1.0), and Fourth Edition of the JBI Reviewer’s Manual, and was registered with PROSPERO (CRD42023460967). Electronic data sources consulted included Central, Medline, Embase, PsycINFO, Scopus, ERIC, and ScienceDirect, using standardized keywords and open-text phrases. The JBI checklist was used to evaluate the quality of case reports and case series.

Results

Thirty-four case reports involving 41 patients were selected. For calcified canals, non-surgical endodontic treatment utilizing chelating agents and flexible rotary instruments has proven highly effective. Furthermore, the employment of a dental operating microscope in conjunction with ultrasonic tips has been reported to improve clinical outcomes in several cases. Also, the use of cone beam computed tomography (CBCT) along with 3D print guides considerably increased success rates in managing these cases.

Conclusions

This review highlights techniques like ultrasonic instruments, flexible nickel-titanium (Ni-Ti) files, and newer methods such as advanced imaging (CBCT) and 3D printed guides. Many case reports show that these approaches lead to higher success rates. Dentists should get familiar with these methods to improve patient outcomes and monitor for any complications. Future research should focus on creating standard protocols and exploring new tools and imaging to advance endodontics.

Introduction

Calcified root canals are a frequent challenge in endodontics. They occur when hard tissue builds up in the canals, leading to partial or complete loss of the canal space. Calcified metamorphosis or pulp canal obliteration is defined as pulpal response to trauma characterized by rapid deposition of hard tissue within the canal space; entire space may appear obliterated radiographically due to extensive deposition, even though some portion of the pulp space remains in histological sections (American Association of Endodontists, 2020). This condition affects approximately 4–24% of teeth requiring root canal treatment (Doranala et al., 2020). They occur when minerals such as calcium salts accumulate within the root canal space, hardening dentin and blocking the passage (Maia et al., 2019; Gürsoy Emek et al., 2022). Calcification can occur either partially or completely, which complicates the process of locating, cleaning, and shaping the canals necessary for effective disinfection and filling (Moradi Majd et al., 2014).

Why do these calcifications occur? Calcifications in root canals may develop from aging, trauma, inflammation, or other pathological factors (Ishak et al., 2020; Heptania, Farahanny & Abidin, 2022; Lewis & Aggarwal, 2023). Dental injuries resulting from substantial force may cause internal bleeding in the pulp cavity and subsequent inflammation which leads to deposition of minerals in the canal. These cases are mostly seen in the anterior teeth of young adults (Chaudhary & Chaudhary, 2022). As age advances, there is an increase in the amount of deposition of secondary dentin, which may contribute to calcification (Freire et al., 2021). Conditions like pulp stones or pulpal sclerosis can also cause calcific material deposition. Also, the presence of deep caries or previous procedures like pulp capping can introduce irritants or materials that promote hard tissue formation (Majid & Elemam, 2015; Yang et al., 2016; Nazar, George & Mathew, 2022).

Locating a calcified root canal during the preparation of the access cavity leads to a variety of challenges, mainly because of limited visibility and variations in the anatomy of the canal (Kumar et al., 2022). Effectively managing these cases is key to achieving successful endodontic outcomes. Failing to address them can lead to persistent infections, treatment failure, and possibly tooth loss (Freire et al., 2021; Gürsoy Emek et al., 2022). Restricted space within the root canal can lead to an increased risk of fracture of both hand and rotary instruments. It also makes it difficult to achieve the required biomechanical preparation (Almohaimede, 2018; Doranala et al., 2020).

Proper management of these cases is crucial for relieving symptoms, ensuring long-term dental health, and improving patient comfort. By using newer techniques, treatment outcomes can be enhanced, and the risk of complications can be reduced.

Treatment of cases with canal obliteration is discussed in many case reports. With the careful use of hand files, chelating agents like ethylenediaminetetraacetic acid (EDTA), and proper irrigation using sodium hypochlorite, it is often possible to successfully negotiate even heavily calcified root canals. The use of dental operating microscopes has been shown to improve accuracy during procedures and may lead to better treatment outcomes (Doranala et al., 2020). Step-back preparation reduces risks like perforation or ledging of the canal. Rotary nickel-titanium (Ni-Ti) systems proved to be valuable in the mechanical shaping of root canal space (Almohaimede, 2018). For successful treatment outcomes, it is essential to plan the treatment meticulously along with the use of cone beam computed tomography, which can provide detailed insights into anatomy such as the extent and location of calcification, and helps in identifying the position of the calcified canal by utilizing 3D print guides (Freire et al., 2021).

Various management techniques are described in the literature for managing these cases but still, it lacks a clear standardized approach (Raghuvanshi, Jain & Kapadia, 2015). Case reports provide great details about techniques for managing included cases, but they are not frequently reviewed systematically. A thorough exploration of these case reports will help to review the most successful techniques which can provide an evidence-based guide for clinicians and also it can point out areas that require further study.

This review aims to summarize techniques mentioned in case reports for managing calcified canals, focusing on effective methods, clinical outcomes, success rates, and common complications. It addresses three key questions: What are the most effective techniques?

What are the outcomes and success rates?

How are common complications managed?

This review will systematically compile and analyze case reports on managing calcified canals, focusing on methods, outcomes, and associated complications. It aims to provide clinicians with effective strategies, improve treatment outcomes, and highlight research gaps for future advancements in endodontics.

Materials and Methods

A thorough systematic review of case reports was conducted. Research was accredited with PROSPERO database under reference number CRD42023460967 and adhered to guidelines set out by PRISMA 2020, Cochrane Handbook for Systematic Reviews of Interventions, version 5.1.0, and JBI Reviewer’s Manual, 4th Edition (Hilton, 2024; Page et al., 2021).

Eligibility criteria

Inclusion criteria

(a) Population: Studies involving the adult population undergoing root canal treatment.

(b) Exposure: Studies including participants with calcified root canals irrespective of the teeth affected, reason for calcification, or treatment performed. Cases were included based on radiographic signs of canal obliteration consistent with the AAE definition of calcific metamorphosis as a pulpal response involving extensive hard tissue deposition within the canal space (American Association of Endodontists, 2020).

(c) Outcome: Studies giving information about different treatment methods used for the management of calcified root canals.

(d) Study design: i. Publications in any language with an available English translation

ii. Research published till 30-04-2024.

iii. Case reports and case series were included.

iv. Research that included full-text articles.

Exclusion criteria

(a) Research studies are not entirely accessible in the database.

(b) The studies only provide abstracts, not the whole texts.

(c) Studies not mentioning required outcomes were excluded.

(d) Clinical studies: Excluded because they often lack detailed case-specific data on treatment of calcified canals, focusing instead on broader treatment outcomes.

(e) Cross-sectional studies: Excluded as they typically analyze prevalence or associations rather than specific management techniques or treatment outcomes for calcified canals.

(f) In vitro studies: Excluded because they do not involve clinical scenarios or patient outcomes, focusing instead on laboratory-based evaluations of materials or techniques.

Search strategy

The Population, Exposure, Outcome, and Study Design (PEOS) inclusion criteria of the review protocol were utilized to systematically identify pertinent studies. Two independent reviewers, K.G. and A.M.P., conducted a comprehensive screening of titles and abstracts to identify potentially eligible articles. Discrepancies encountered during the selection process were initially addressed through structured discussions aimed at reaching consensus, based on a re-evaluation of the predefined inclusion and exclusion criteria. In cases where consensus could not be achieved, a third senior reviewer, (K.S.B.), was consulted to serve as the adjudicator. His role ensured impartial arbitration, thereby maintaining objectivity and methodological rigor in the study selection process.

For the detailed search, we used Medline, CINAHL, Embase, PsycINFO, Scopus, ERIC, ScienceDirect, and the Cochrane Central Register of Controlled Trials (Central) using controlled vocabulary and free text keywords (Refer to Tables 1 and 2).

Table 1 Search strategy according to PEOS.

	Search strategy	
Population	(“dental pulp cavity”[MeSH Terms] OR (“dental”[All Fields] AND “pulp”[All Fields] AND “cavity”[All Fields]) OR “dental pulp cavity”[All Fields] OR (“root”[All Fields] AND “canal”[All Fields]) OR “root canal”[All Fields]) AND (“therapy”[Subheading] OR “therapy”[All Fields] OR “treatment”[All Fields] OR “therapeutics”[MeSH Terms] OR “therapeutics”[All Fields])	
Exposure	(((“dental pulp cavity”[MeSH Terms] OR (“dental[All Fields] AND “pulp”[All Fields] AND “cavity”[All Fields]) OR “dental pulp cavity”[All Fields] OR (“root”[All Fields] AND “canal”[All Fields]) OR “root canal”[All Fields]) AND (“calcinosis”[MeSH Terms] OR “calcinosis”[All Fields] OR “calcification”[All Fields] OR “calcification, physiologic”[MeSH Terms] OR (“calcification”[All Fields] AND “physiologic”[All Fields]) OR “physiologic calcification”[All Fields])) AND (“organization and administration”[MeSH Terms] OR (“organization”[All Fields] AND “administration”[All Fields]) OR “organization and administration”[All Fields] OR “management”[All Fields] OR “disease management”[MeSH Terms] OR (“disease”[All Fields] AND “management”[All Fields]) OR “disease management”[All Fields])) AND (“case reports”[All Fields] OR “case report”[All Fields])	
Study design	“Case reports” [All Fields] OR “case report" [All Fields]	

Table 2 Overall search strategy in PubMed.

Search strategy	No. of articles obtained	
(((((“dental pulp cavity”[MeSH Terms] OR (“dental”[All Fields] AND “pulp”[All Fields] AND “cavity”[All Fields]) OR “dental pulp cavity”[All Fields] OR (“root”[All Fields] AND “canal”[All Fields]) OR “root canal”[All Fields]) AND (“calcinosis”[MeSH Terms] OR “calcinosis”[All Fields] OR “calcification”[All Fields] OR “calcification, physiologic”[MeSH Terms] OR (“calcification”[All Fields] AND “physiologic”[All Fields]) OR “physiologic calcification”[All Fields])) OR ((“dental pulp cavity”[MeSH Terms] OR (“dental”[All Fields] AND “pulp”[All Fields] AND “cavity”[All Fields]) OR “dental pulp cavity”[All Fields] OR (“pulp”[All Fields] AND “canal”[All Fields]) OR “pulp canal"[All Fields]) AND obliteration[All Fields])) OR ((“dental pulp cavity"[MeSH Terms] OR (“dental"[All Fields] AND “pulp"[All Fields] AND “cavity"[All Fields]) OR “dental pulp cavity”[All Fields] OR (“pulp”[All Fields] AND “canal”[All Fields]) OR “pulp canal”[All Fields]) AND (“calcinosis”[MeSH Terms] OR “calcinosis”[All Fields] OR “calcification”[All Fields] OR “calcification, physiologic”[MeSH Terms] OR (“calcification”[All Fields] AND “physiologic”[All Fields]) OR “physiologic calcification”[All Fields]))) AND “management”[All Fields] OR “disease management”[MeSH Terms] OR (“disease”[All Fields] AND “treatment”[All Fields]) OR “disease treatment”[All Fields])) AND (“case reports”[All Fields] OR “case report”[All Fields])	1,489	

We searched all articles published up to April 30, 2024, regardless of the language of the publication.

Focused review question

Which approaches are used for the management of calcified canals during root canal treatment?

Selection of studies

Two separate reviewers looked over every research study’s title and abstract and gave them their honest opinions. In order to implement selection criteria, the following procedures were followed: (a) Combining the search results to remove duplicate items.

(b) Sorting out irrelevant articles by reading their abstracts and titles.

(c) Retrievement of whole articles that may include vital information.

(d) Putting together and binding many publications based on the same study.

(e) Reviewing the whole texts of the publications to see how well the research met the qualifying requirements.

(f) Connecting with researchers to determine study eligibility if needed.

(g) Picking to go on with data collecting and include the research.

Data extraction

For each of the listed studies, two reviewers independently extracted data. Discussion once again led to the resolution of disagreements. The checklist of factors was used to guide the data extraction process

Here are the key points from this inventory: Title, year, and author details

Region

Research design

Demographic age group

Sex

Symptoms and signs

Radiographic features

Outcomes

Treatment performed

Conclusion and other items

Careful and exact data was gathered from all selected studies regarding the study and publication details, study population, research environment, treatments, control groups, evaluation criteria, methodology, data analysis, and findings. Any additional relevant information, such as financial assistance, conflicts of interest, etc., was also diligently retrieved. All of the data extraction for the important outcomes was done and meticulously documented in Excel spreadsheets.

Results

Study selection

The first electronic database exploration resulted in 5,109 titles from PubMed/Medline, the Cochrane Library, and Directory of Open Access Journals (DOAJ). There were 38,832 items that were referenced twice. Two separate reviewers looked over abstracts and chose 1,271 titles that were relevant; 517 titles were deemed unrelated to the subject and were subsequently deleted. Altogether 126 articles were chosen for complete article assessment after reviewers had examined and discussed them. Manually searching reference lists of chosen studies did not turn up any further publications. Only 34 studies made it through pre-screening, inclusion/exclusion criteria, and PEOS question processing. Data extraction and statistical analysis were performed on 34 studies that were incorporated in the qualitative synthesis (Fig. 1).

Figure 1 The PRISMA 2020 flow diagram.

Study characteristics

Thirty-four case reports were included in this systematic review, whose general characteristics are mentioned in Table 3. These cases were identified in different parts of the world India (Gopikrishna, Parameswaran & Kandaswamy, 2004; Koli et al., 2014; Raghuvanshi, Jain & Kapadia, 2015; Hegde et al., 2019; Doranala et al., 2020; Sinha et al., 2022; Nazar, George & Mathew, 2022; Kumar et al., 2022; Chaudhary & Chaudhary, 2022; Lewis & Aggarwal, 2023; Panithini et al., 2023), Germany (Krastl et al., 2016; Tchorz, Wrbas & Hellwig, 2019), Libya (Majid & Elemam, 2015), China (Shi et al., 2017), Saudi Arabia (Almohaimede, 2018), Brazil (Fonseca Tavares et al., 2018; Lara-Mendes et al., 2018; Maia et al., 2019; Tavares et al., 2020; Freire et al., 2021; Gonçalves et al., 2021; Santiago et al., 2022), Belgium (Torres et al., 2018), Lebanon (Ishak et al., 2020), Pakistan (Irfan et al., 2020), Portugal (Quaresma et al., 2022), Japan (Takeichi, 2021; Sekiguchi, 2022), Turkey (Gürsoy Emek et al., 2022), Indonesia (Heptania, Farahanny & Abidin, 2022), Iran (Nabavi, Navabi & Mohammadi, 2022; Zargar & Amiri, 2023), Sri Lanka (Palipana, 2023). The majority of cases were reported in India, followed by Brazil. The most common reason for patient complaints was previous trauma, parafunctional habits like bruxism, dental caries, orthodontic treatment, prosthodontic concerns, etc. Patients’ ages varied from 15–70. Some of the case reports included cone beam computed tomography (CBCT) examination to assess the extent of calcification of root canals. Maxillary and mandibular anterior teeth were commonly affected.

Table 3 Study characteristics.

Study ID	Place of study	Age/Gender	Chief complaint	History	Teeth affected	Clinical examination	Radiographic examination	CBCT findings	Treatment planned	Follow-up	Author conclusions	
Gopikrishna, Parameswaran & Kandaswamy (2004)	India	21/M	Swelling and discharge with upper anterior.	Trauma 9 years back	11,21	Diffuse swelling, sinus discharge, and positive pain on percussion. A negative response to EPT	Completely obliterated pulp chamber and pulp canal	–	Nonsurgical endodontic treatment	1 year	In cases with such calcified chambers and canals, the use of chelating agents EDTA and NaOCl is highly useful	
Koli et al. (2014)	India	24/M	Persistent suppuration in the maxillary anterior region for the last 6 months.	Blunt trauma	21,11,12	No tenderness on percussion, negative vitality tests	A well-defined radiolucency with a radiopaque margin, caused by an augmented trabecular pattern, present at the apices of 12 and 11	–	Nonsurgical endodontic treatment	Not mentioned	The use of ultrasonics in endodontics presents several benefits, such as better visualization and a more conservative approach for selective tooth structure removal.	
Krastl et al. (2016)	Germany	15/M	Pain in maxillary right central incisor	Trauma 7 years ago	11	Discoloration, tenderness on percussion, and an absence of response to both cold and electric pulp tests were noted.	Complete loss of pulp chamber and canal space	The CBCT showed clear signs of apical periodontitis.	Guided endodontic surgery using a 3D template	15 months	The use of guided endodontics provides a reliable and minimally invasive solution for locating calcifications and reducing the risk of perforation in cases where standard endodontic access is difficult.	
Majid & Elemam (2015)	Libya	17/F	Persistent, sharp, and spontaneous discomfort in the upper right back teeth region for the last three days.	Not mentioned	14	No tenderness on percussion, negative vitality tests	There was minimal widening of the periodontal ligament space. No detectable periapical pathology. A deep radiopaque filling was encircled by a radiolucent region suggestive of secondary caries. The apical portion of the root canal system seems calcified and exhibited a distal curvature.	–	Nonsurgical endodontic treatment	3 weeks	Root canal treatment prognosis in these cases is determined by the ongoing health of the pulp or periapical area beyond the obstruction. In absence of clinical symptoms or apical disease, the treatment outcome remains positive.	
Raghuvanshi, Jain & Kapadia (2015)	India	26/M	Intermittent pain localized to the upper anterior jaw for a time period of 1 month.	Trauma 5 years ago	21	Discoloration and intro-oral sinus, nontender to vertical and horizontal percussion. Thermal and electric pulp response was negative	Pulp canal obliteration with 21#	–	Nonsurgical endodontic treatment	Not mentioned	The cases presented are the example of radiographically unidentifiable pulp chamber that is difficult to treat but manageable via nonsurgical root canal therapy	
21/M	Discoloration in the upper front region of the jaw since 6 months	Trauma 2 years back	11	Discoloration and intro-oral sinus, nontender to vertical and horizontal percussion. Thermal and electric pulp response was negative	Pulp canal obliteration with 11	–	Nonsurgical endodontic treatment	Not mentioned		
Shi et al. (2017)	China	29/F	Pulpal calcification detected during the endodontic procedure for the lower right first molar.	Not mentioned	46	The tooth was sensitive to percussion but did not react to thermal or electric pulp stimulation	–	–	Guided endodontic surgery	6 months	Guided endodontics provides a safe and efficient method for addressing pulp canal calcification and apical pathology. The use of 3D-printed templates enhances canal localization, ensuring a more accurate and predictable treatment.	
Almohaimede (2018)	Saudi Arabia	17/F	Pain of mild to moderate severity localized in mandibular right posterior teeth and aggravated by mastication.	Not mentioned	46	The percussion test showed mild to moderate sensitivity, and the palpation test was non-tender.	IOPA-A radiolucent periapical lesion was present. Narrow mesial canals and restorative material extending into a partially calcified pulp chamber.	Mesiolingual (ML) canal was completely calcified	Nonsurgical endodontic treatment	Not mentioned	Effective management of calcified pulp chambers and canals requires the use of multiple supplementary tools. CBCT should not replace traditional radiography but should be used when its advantages and indications justify its use.	
Lara-Mendes et al. (2018)	Brazil	26	Clinical signs present in the area of the upper central incisors.	Trauma	11,21	The tooth did not react to thermal or electric pulp testing, but the patient reported tenderness on percussion, consistent with early acute apical periodontitis.	–	An apical radiolucency was noted in the left maxillary central incisor, matching the patient’s pain description. The canal was visible only in the apical 2 mm of the root.	Guided endodontic surgery	1 year	A case report is described, which highlights the treatment of a severely calcified maxillary central incisor, with modifications in access design to reduce incisal edge	
Fonseca Tavares et al. (2018)	Brazil	43/F	History of pain in the maxillary right central incisor	Trauma 25 years back	11	The tooth was discolored and yellow, and it presented tenderness to percussion. Thermal and electrical sensitivity tests were negative	Severely calcified root canal	–	Guided endodontic surgery	15 days	The conventional opening access programming of guided endodontics in cases of PCC in anterior teeth with apical periodontitis is very reliable and permits proper root canal disinfection expeditiously	
24/F	Endodontic treatment of the maxillary right central incisor	Trauma	11	Sensitive to percussion and palpation tests.	–	Confirmed the presence of severe canal calcification and apical periodontitis.	Guided endodontic surgery	30 days	
Torres et al. (2018)	Belgium	85/F	Root canal treatment	Not mentioned	22	Absence of percussion tenderness and no evidence of a sinus tract.	–	A calcified root canal was observed up to the apical third along with an apical radiolucent area.	Guided endodontic surgery	6 months	Utilizing the Microguided Endodontics technique, a minimally invasive access was established up to the mid-root in a maxillary lateral incisor with PCO and apical periodontitis. This approach helps in negotiating canal obliteration efficiently while reducing both chair time and the risk of iatrogenic complications.	
Hegde et al. (2019)	India	24/M	Discoloration of teeth	Trauma	11	Negative electric pulp test	Radiographic findings revealed a fully calcified root canal in tooth 11 and a periapical radiolucent area.	Assessment showed that the root canal was obstructed in the cervical and middle thirds. The apical 7 mm was still accessible.	Guided endodontic surgery	Not mentioned	In this case, guided endodontic therapy facilitated a more efficient management of obliteration in anterior teeth, which reduces treatment time.	
Maia et al. (2019)	Brazil	47/F	Presence of acute symptoms	Not mentioned	26	Sensitivity to percussion was noted, with a negative reaction to both thermal and electrical stimuli.	Radiographic findings suggested an obliterated pulp chamber and possible calcification of the buccal canals.	–	3D printed endodontic guide	15 days. 6,12 months	These case reports highlight how technological advancements can facilitate the broader implementation of guided endodontic techniques, making them more accessible to clinicians with varying levels of experience.	
65/F	A complaint of pain in the upper left second premolar.	Parafunctional habits	25	Clinical examination showed palatal cusp fracture and tenderness on percussion.	–	Confirmed pulp obliteration	
45/F	Endodontic treatment	Not mentioned	15	Apical periodontitis and positive response to percussion and palpation tests.	–	Radiographic analysis showed two severely calcified root canals joining at a single foramen in the apical third.	
Tchorz, Wrbas & Hellwig (2019)	Germany	42/F	Minor symptoms of tenderness to percussion in the mandibular left central incisor	Not mentioned	31	No tenderness on percussion, negative vitality tests	The root canal exhibited calcification up to the apical third and periapical inflammation.	–	Guided endodontic surgery	Not mentioned	The utilization of an access guide for the localization of remaining pulp spaces is a reliable method in cases of PCC	
Doranala et al. (2020)	India	51/M	A 1-month history of pain localized to the maxillary central incisor.	Trauma 3 years back	11	The tooth presented with an intact crown, tenderness to percussion, and no evidence of mobility.	The coronal aspect of the tooth exhibited increased radiopacity, with pulp canal obliteration and an adjacent radiolucent lesion.	Evidence of partial calcification was noted in the coronal aspect, approaching the cervical third of the root.	3D printed endodontic guide	3 months	Guided endodontics has demonstrated effectiveness in creating minimal access cavities along with predictable outcomes in cases of calcific metamorphosis.	
Freire et al. (2021)	Brazil	23/F	Discoloration affecting the crown of the upper left central incisor was noted.	Trauma	21	Tenderness on percussion was noted, with an absence of response to pulp vitality testing.	Calcification of the pulp and canal was evident in the IOPA radiograph.	A reduction in pulp canal space was observed	Guided endodontic surgery	2 years	Guided endodontics facilitates both the planning and execution phases, assisting clinicians in treating calcified teeth.	
Ishak et al. (2020)	Lebanon	52/F	Dull discomfort while chewing.	Bruxism	31,41	Sensibility assessment demonstrated a positive response to percussion.	Significant calcification affecting the pulp chamber and root canal, without any detectable radiolucent areas.	Radiographic evaluation showed canal visibility beyond the coronal-middle third junction, appearing at 9 mm for tooth 31 and 4 mm for tooth 41.	Guided endodontic treatment	Not mentioned	The use of an endodontic guide during the treatment of calcified canals may improve outcomes and alleviate stress for both the clinician and the patient.	
Irfan et al. (2020)	Pakistan	42/M	A 2-week history of biting pain in one of the upper left teeth.	Caries	24	Tooth tender to percussion, negative response to pulp sensitivity tests	The absence of visible root canal outlines on the periapical radiograph suggested calcification, accompanied by a periapical radiolucent area.	Compared to other teeth, the canal image was less distinct. The buccal canal was absent in the axial view at all root levels.	Nonsurgical endodontic treatment	Not mentioned	With the use of proper instruments and advanced imaging techniques like CBCT, treating calcified canals is possible. While 2D radiographs at different angles contribute to diagnosis, CBCT plays a pivotal role by offering a more detailed view of the root canal anatomy.	
Tavares et al. (2020)	Brazil	35/F	Pain	Not mentioned		The patient reported pain on palpation along with percussion sensitivity in the affected tooth.	A completely obliterated root canal space was observed, along with an apical radiolucency.	confirmed severe PCO	Guided endodontic treatment + antimicrobial photodynamic therapy (aPDT).	12 months	At the 1-year follow-up, the combined approach of guided endodontics and aPDT proved successful in treating severe PCO and apical periodontitis, as evidenced by clinical, radiographic, and tomographic assessments.	
Gonçalves et al. (2021)	Brazil	40/F	Pain in the upper left canine region	Orthodontic treatment	23	Slight discoloration of the crown, negative pulp test, absence of pain	–	Severe calcification of root canal up to apical third	Guided endodontic treatment	60 days	The use of guided endodontics allowed the preservation of a large part of the dental structure.	
Kumar et al. (2022)	India	35/F	Darkening of the tooth in the upper front region of the jaw for the past 6 months	Trauma	21	No tenderness on percussion, Thermal and electric pulp testing showed a negative response	The outline of the root canal was not clear on the periapical radiograph which suggested the calcification of the root canal.	The canal was not visible till the middle third of the root canal	Nonsurgical endodontic treatment	Not mentioned	With the application of CBCT as a guide, it is helpful in orienting the burs angulation within the pulp chamber with fewer chances of iatrogenic injury	
42/M	Discoloration of a tooth in the upper front tooth region of the jaw for the past 2 months	Trauma	21	No tenderness on percussion, Thermal and electric pulp testing showed a negative response	The outline of the root canal was not clear on the periapical radiograph which suggested the calcification of the root canal.	The canal was not visible till the middle third of the root canal	Nonsurgical endodontic treatment	Not mentioned	
Quaresma et al. (2022)	Portugal	58/F	Sporadic discomfort imprecisely localized in the right maxillary region, which worsened when chewing, and had been present for several months	Not mentioned	16	Grade 1 tooth mobility, positive complaints in response to percussion and palpation tests, and no response to the cold pulp test	Alterations at the level of the periapical bone trabeculae and an apparent reduction in the volume of the pulp chamber	Severe calcification of the first mesiobuccal (MB1) canal was evident	Nonsurgical endodontic treatment	Not mentioned	The application of CBCT at various intervals during endodontic treatment allowed three-dimensional localization of the anatomical variations, and location of the canal orifice.	
32/F	Endodontic evaluation	Orthodontic treatment	41	A mild degree of mobility (Grade 1) was observed, with periodontal probing values within the normal range. The tooth was non-tender to percussion but showed an increased pain response to cold testing.	Severe root canal calcification, presence of internal resorption	Marked calcification was noted within the root canal, accompanied by internal resorption displaying an irregular expansion pattern in the coronal third, with no signs of perforation.	Nonsurgical endodontic treatment	Not mentioned	
Takeichi (2021)	Japan	70/F	Slight occlusal pain in the tooth	Not mentioned	26	Percussion testing resulted in mild to moderate tenderness, with no pain detected during palpation.	Structural changes at the periapical bone level were evident, along with a reduced pulp chamber volume.	The root canals were indistinct, except for the middle third, which was partially visible.	Nonsurgical endodontic treatment	Not mentioned	Managing pulp canal obliterations necessitates CBCT interpretation, the aid of a surgical microscope, and the use of Ni-Ti instruments.	
Chaudhary & Chaudhary (2022)	India	28/F	Persistent pain in the maxillary anterior region for the past month.	Trauma 2 years back	21	The crown showed mild discoloration, with tenderness upon palpation and percussion, and no response to cold or electric pulp sensitivity tests.	Obliteration of the pulp canal and widening of the periodontal ligament space were observed on the IOPA.	–	Nonsurgical endodontic treatment	1 week	This case report highlights that while managing calcified canals is challenging, the successful outcome was achieved through a thorough understanding of tooth morphology, proper canal negotiation techniques, appropriate use of instruments, materials, and diagnostic aids, along with patience, effort, and care.	
Gürsoy Emek et al. (2022)	Turkey	22/F	Pain in the maxillary left second premolar teeth	Orthodontic treatment	25	Lack of response to thermal and electrical sensitivity tests, along with intense percussion sensitivity during bite pressure.	Obliterated pulp chamber and calcified root canal.	–	3D printed endodontic guide	7 days, 1 month, 3 months	By using guided endodontics (GE) and 3D printing technology, a secure approach to treating obliterated root canals can help prevent various complications.	
Heptania, Farahanny & Abidin (2022)	Indonesia	20/F	Had been experiencing a cavity in the lower left molar for the past seven months.	Caries	36	The electric pulp test gave a positive response, and there was no sensitivity to percussion or palpation.	The X-ray revealed overlapping mesial root canals, which appeared as two calcified mesial roots.	–	Nonsurgical endodontic treatment	Not mentioned	Endodontists often face challenging cases like the one presented, where success depends on clinical knowledge, magnification tools, and ultrasonic tips.	
Nabavi, Navabi & Mohammadi (2022)	Iran	58/M	Root canal therapy was carried out on teeth #24, #25, and #26 for prosthodontic reasons.	Prosthodontic reason	24,25,26	The patient showed no symptoms, with no pain on percussion and no sinus tract.	The anterior teeth exhibited partial canal obliteration.	Calcification was observed in the root canal within the coronal third of the root.	3D printed endodontic guide	6 months	The coronal third of the root exhibited calcification in the root canal.	
Nazar, George & Mathew (2022)	India	48/F	The upper left front tooth area has experienced swelling for the past 8 months.	Orthodontic treatment	22	Well-defined swelling with a diameter of 1 cm in relation to the upper left lateral incisor, tender on percussion, and a negative response to EPT	A round, clearly defined periapical radiolucency was seen, with total obliteration of the pulp chamber and canal in the coronal section.	Periapical lesion having a diameter of 8.3 mm breaching the labial cortical plate	3D printed endodontic guide	1,2,3 months	The findings of this case report highlight that large periapical lesions can have favorable outcomes with nonsurgical treatment, which should always be considered the first treatment option.	
Santiago et al. (2022)	Brazil	58/F	Endodontic treatment of the frst right mandibular molar	Caries	44	The patient did not experience any symptoms and had negative responses to both thermal and percussion tests.	The mesial and distal roots exhibited complete canal obliteration	The mesiobuccal (MB) and mesiolingual (ML) canals were found to be completely obliterated.	Guided endodontics	Not mentioned	Guided endodontics offers a personalized approach that enhances safety, minimizes the risk of root perforation, and significantly reduces treatment time.	
Sekiguchi (2022)	Japan	40/F	Severe swelling and pain 10 days ago	Orthodontic treatment	43,13	Negative responses to thermal and electrical tests, along with severe percussion sensitivity when biting, were observed.	The root canal showed signs of calcification, and the pulp chamber was obliterated.	No pulp cavity was apparent	Microendosurgery	Not mentioned	Treatment was undertaken using microendosurgery since access from the crown side was not possible. As a result the patient has progressed with a good prognosis	
Sinha et al. (2022)	India	64/M	A history of pain in tooth 14 has been reported for the past month.	Caries	14	Mesial proximal caries was found, and the tooth exhibited extreme sensitivity to percussion.	Two roots were identified, with the pulp chamber and root canals completely obliterated.	–	Nonsurgical endodontic treatment	Not mentioned	Dealing with teeth affected by PCO requires a significant investment of time and effort, yet combining new technologies, a comprehensive understanding of pulp anatomy, skilled radiographic techniques, and patience ensures successful outcomes.	
18/F	Pain in the lower right back tooth has been present for the last month.	Orthodontic treatment	46	Periodontal pockets were found on both the mesial and distal surfaces of tooth 46, and the tooth was tender to percussion.	The pulp chamber was completely obliterated, while the root canals were still patent.	–	Nonsurgical endodontic treatment	Not mentioned	
Lewis & Aggarwal (2023)	India	19/F	There was pain on bite pressure in the maxillary anterior region.	Trauma	21,22	Tooth #21 exhibited severe destruction and was submerged within the gingiva and alveolar socket, while tooth #22 had a restoration in place.	–	There was obliteration of the pulp chamber, accompanied by defined periapical radiolucency parameters.	Nonsurgical endodontic treatment	1,3,6,12 months	By employing a minimally invasive method, guided endodontics successfully addresses pulp and periapical diseases in teeth.	
Palipana (2023)	Sri Lanka	26/F	Tenderness around the apex of the tooth	Orthodontic treatment	21	Positive on tenderness to percussion	–	–	Nonsurgical endodontic treatment	1 month	CBCT helps in determining the most minimally invasive technique for negotiating teeth with PCC, resulting in a predictable treatment outcome.	
Panithini et al. (2023)	India	35/F	The upper maxillary central incisor shows signs of discoloration	Trauma	21	Tooth tender to percussion, negative response to pulp sensitivity tests	The tooth shows an obliterated pulp chamber, calcified root canal, and a periapical radiolucent area.	–	Nonsurgical endodontic treatment	Not mentioned	Dynamic navigation enabled the precise formation of a minimally invasive access path in this case, with real-time feedback provided by the drilling device.	
Zargar & Amiri (2023)	Iran	32/F	A discoloration was observed in the left maxillary central incisor.	Trauma 10 years back	21	Discoloration, No response was recorded for the pulp sensibility tests, and percussion and palpation revealed no symptoms. ration,	Pulp canal calcification and rarefaction in the periapical area	A 15 mm obliteration was observed in the coronal and middle thirds of the canal.	Guided endodontic treatment	18 months	This case highlights the use of RCT with an endodontic guide for a calcified maxillary central incisor, where access was adjusted to preserve the incisal edge, contributing to a more favorable long-term prognosis.	

Risk of bias assessment

It is presented in Fig. 2.

Figure 2 Risk of Bias assessment (Doranala et al., 2020; Gürsoy Emek et al., 2022; Maia et al., 2019; Heptania, Farahanny & Abidin, 2022; Ishak et al., 2020; Lewis & Aggarwal, 2023; Chaudhary & Chaudhary 2022; Freire et al., 2021; Nazar, George & Mathew, 2022; Majid & Elemam, 2015; Kumar et al., 2022; Almohaimede, 2018; Takeichi, 2021; Irfan et al., 2020; Raghuvanshi, Jain & Kapadia, 2015; Koli et al., 2014; Gopikrishna, Parameswaran & Kandaswamy, 2004; Nabavi, Navabi & Mohammadi, 2022; Sinha et al., 2022; Hegde et al., 2019; Tchorz, Wrbas & Hellwig, 2019; Tavares et al., 2020; Santiago et al., 2022; Quaresma et al., 2022; Zargar & Amiri, 2023; Palipana, 2023; Lara-Mendes et al., 2018; Shi et al., 2017; Fonseca Tavares et al., 2018; Torres et al., 2018; Gonçalves et al., 2021; Krastl et al., 2016; Panithini et al., 2023; Sekiguchi, 2022).

Consistency in reporting

All studies consistently provided information regarding demographic characteristics, clinical presentation, screening tests used, treatment, post-operative condition, and takeaway lessons. While some reports did not utilize CBCT, they still provided information through detailed clinical and radiographic records. However, it is important to note that CBCT offers more comprehensive three-dimensional information, which may enhance documentation quality in complex cases.

While multiple reports claim they successfully treated calcified canals without CBCT, the inconsistent utilization of this advanced imaging technology may lead to treatment planning bias. CBCT provides a better visualization of canal anatomy and the extent of canal calcification, which can affect treatment decisions and their respective outcomes. The studies which relied on conventional 2D imaging alone may have either had a much more difficult time negotiating the canal or were incomplete assessments. Outcomes amongst these reports may not be entirely verifiable, and the lack of CBCT imaging on occasion may have actually understated the procedural complexity or diagnostic certainty.

Variability in reporting patient history

There is significant variability in the reporting of patient history. Several studies were marked as “Unclear,” indicating that the patient history was either not well-documented or was insufficiently detailed. It is a significant limitation because understanding a patient’s history is important for the interpretation of outcomes of provided treatment.

Lack of reported complications

Out of 34 case reports, none of them reported adverse effects, which is unusual. This suggests that there could be underreporting or an actual lack of adverse events.

Overall appraisal

A total of 27 out of 34 studies were rated as ‘Low’, which is likely due to gaps in reporting, especially regarding patient history and the lack of adverse event documentation. Only seven studies were rated as “Moderate,” which suggests that they had more thorough reporting or a stronger methodological approach.

The overall quality of the studies, indicated by the “Low” appraisal ratings for most, suggests that while they were generally strong in areas like clinical presentation and treatment descriptions, they lacked thorough documentation of patient history and adverse event reporting. These gaps may limit the reliability and applicability of the findings, which highlight the need for more rigorous and comprehensive reporting standards in future research.

Discussion

Managing cases of canal calcifications is one of the most challenging aspects of endodontic treatment due to the obstruction of the canal space, which can affect the success rates. This review analyzes various case reports and highlights key strategies and advanced techniques that have shown promise in overcoming these challenges.

The reports showed that conventional techniques are valuable, but they have notable limitations during the management of these cases. Traditional methods, particularly the use of rigid hand instruments (K files and C+ files) and 2D radiographic imaging, do not often work well when managing the complicated and narrow anatomy found in calcified root canals. These instruments cannot provide great flexibility and tactile sensitivity and expose operator error during these difficult situations. These conventional procedures may increase the risks of procedural errors, including instrument fractures, ledges, transportation, and canal perforations (Doranala et al., 2020; Nabavi, Navabi & Mohammadi, 2022). In addition to that, conventional radiography provides two-dimensional images that may not accurately represent the three-dimensional structure of the canal, which complicates diagnosis and further treatment.

However, advancements in endodontic instruments and techniques have significantly improved the predictability of outcomes, even in complex cases. Many case reports stated that the use of Ni-Ti files in these cases is beneficial with fewer procedural errors (Gopikrishna, Parameswaran & Kandaswamy, 2004; Raghuvanshi, Jain & Kapadia, 2015). Also, the introduction of file systems like continuous rotary systems like Pro-Taper and reciprocating single file system such as Wave One consistently showed improved outcomes while maintaining original canal curvatures. These results may be attributed to its properties like flexibility, shape memory, and superior cutting efficiency (Hegde et al., 2019; Nabavi, Navabi & Mohammadi, 2022).

In cases with apical pathologies, inclusion of calcium hydroxide as an interappointment medicament has been identified as beneficial in promoting healing (Majid & Elemam, 2015; Raghuvanshi, Jain & Kapadia, 2015). However, as case reports suggest, the role of antimicrobial photodynamic therapy (aPDT) is gaining attention as a complementary method in disinfection, particularly in cases involving resistant bacterial strains like Enterococcus faecalis (Hegde et al., 2019).

The use of devices such as dental operating microscopes (DOM) and magnification loupes allowed for improved diagnosis and canal negotiation in these situations. Magnification and radiance delivered by a dental microscope and magnifying loupes allow the operator to locate the orifices of these canals, which are not discernible by the naked eye.

Cone beam computed tomography serves as a powerful adjunct in the management of these cases as it delivers a three-dimensional view of the tooth structure, unlike 2D radiography. It enables accurate visualization of calcifications, canal variations, and anatomical landmarks and also scales down errors in diagnosis and treatment planning (Doranala et al., 2020; Freire et al., 2021; Gürsoy Emek et al., 2022). Out of the reviewed case reports, most of them show success rates of more than 85 percent with the resolution of symptoms and radiographic signs of periapical healing with follow-up ranging from 1 year to 5 years in most of them. The higher success rates reported may be influenced by early diagnosis and the application of modern techniques, highlighting the potential clinical value of these tools (Kiefner et al., 2017).

With time, many clinicians have preferred the use of ultrasonic tips along with dental microscopes, which improve visualization and accuracy during access preparation. This approach enables clinicians to navigate canal calcifications more efficiently. Ultrasonic tips can selectively remove tooth structure and provide better tactile feedback, which improves access to calcified canal systems (Almohaimede, 2018; Takeichi, 2021). This technique helps lower the chances of making errors during procedures, like creating perforations or unnecessary removal of dentin, which are often seen with traditional methods.

Recent advancements like the introduction of 3D print guides along with dynamic navigation systems improved the accuracy and efficiency of navigating and shaping these canals. These methods can be routinely used in calcification cases and are particularly useful in teeth with complex root canal anatomy. As described earlier, CBCT provides three-dimensional visualization of the canal space, which helps in the fabrication of 3D print guides. Static guides are fabricated using CBCT data and 3D surface scans, allowing for preoperative planning and the creation of customized guides that direct the bur along a predefined path. Due to a lack of adjustability, static guides are not able to change their direction in case of any variation in the canal during procedures (Krastl et al., 2016; Gonçalves et al., 2021). With these printed guides, one can precisely navigate canals with minimal risk of canal perforation or transportation.

On the other hand, dynamic navigation systems allow real-time tracking, which further improves success rates in locating and navigating canals (Lara-Mendes et al., 2018, 2019; Tavares et al., 2020).

Dynamic navigation systems can remarkably allow flexibility during the procedure, which reduces the chances of errors. Because of its ability to provide real-time feedback, which helps in the negotiation of canals more accurately, even less experienced clinicians can use it efficiently. However, certain factors like specialized training, high initial investment, and maintenance costs limit its use (Panithini et al., 2023). As technology progresses and costs become more affordable, dynamic navigation systems will probably see more widespread acceptance in the industry. The incorporation of these principles into everyday practice will boost clinician confidence and competence and also improve patient outcomes by decreasing procedural risk and optimizing treatment efficacy (Lewis & Aggarwal, 2023).

One significant limitation of this systematic review is the mini-level of evidence; case reports are descriptive reports that provide rich clinical information. The included reports exhibited extreme heterogeneity across treatment protocols, diagnostic methods (for example, CBCT or 2D radiographs), demographics, experience of the operator, and length of follow-up. The heterogeneity and small sample sizes, and the frequent absence of information regarding aetiology (e.g., trauma vs. age-related), limits the generalizability of the findings. Additionally, the preponderance of favourable outcomes and the absence of complications raise questions of publication bias and selection bias, contextually shifting success rates, which averaged upwards of 85%. Success rates should be interpreted with caution, as there is little guidance in the form of defined meanings for clinical success or common protocols for follow-up.

Significantly, this evaluation found no research that directly compared techniques in a standardized clinical context, thus limiting our ability to assess relative effectiveness. Despite this evaluation showing trends, e.g., the increased use of CBCT technology, dental operating microscopes, ultrasonic devices, and guided endodontics, there was no possibility of conducting any comparative analysis. Quality clinical studies, e.g., Krastl et al. (2016) and Kiefner et al. (2017), reported the higher accuracy of guided navigation systems compared with traditional methods, particularly in the management of severely calcified canals. These external studies are helpful for context and demonstrate the necessity for a consideration of wider clinical evidence to complement case-based evidence.

For these reasons, our results should not be interpreted as definitive, but hypothesis-generating. The data provides preliminary evidence to indicate support for advanced methods but is not robust enough for more generalized recommendations. Future research should focus on prospective, multicenter studies with sufficient sample sizes, standardized outcome measures, and longer-term follow-up. Randomized controlled trials comparing traditional with contemporary approaches (e.g., guided endodontics, dynamic navigation, antimicrobial photodynamic therapy (aPDT), and newer Ni-Ti file systems) are dependent on understanding each method’s relative effectiveness, safety, costs and patient-related outcomes. Future studies should focus on promoting research innovation in the area of chelating agents, imaging and access pathways (e.g., 3D printing and artificial intelligence guided navigation) and encourage and generate a more comprehensive series of clinical guidance to help dentists predictably manage calcified canals.

Conclusions

This systematic review highlights the effectiveness of advanced tools like CBCT, DOM, and Ni-Ti systems. It also emphasizes the need for a comprehensive and individualized approach when managing calcified canals. The incorporation of these technologies into everyday endodontic practice can enhance treatment success, especially in cases where conventional methods are lacking. Also, recognizing the present limitations of case reports such as their heterogeneity and small sample sizes, is important. Future research should focus on standardized treatment protocols, long-term studies, and exploring new technologies to improve the management of calcified canals. Clinicians need to stay updated on these advancements and carefully consider their pros and cons to ensure the best patient care.

Supplemental Information

Supplemental Information 1 PRISMA checklist.

Supplemental Information 2 Raw data.

Supplemental Information 3 Registration to PROSPERO.

Additional Information and Declarations

Competing Interests

Ajinkya M. Pawar is an Academic Editor for PeerJ.

Author Contributions

Kunal Giri conceived and designed the experiments, performed the experiments, analyzed the data, authored or reviewed drafts of the article, and approved the final draft.

Kulvinder Banga conceived and designed the experiments, performed the experiments, authored or reviewed drafts of the article, and approved the final draft.

Suraj Arora conceived and designed the experiments, performed the experiments, authored or reviewed drafts of the article, and approved the final draft.

Firas Elmsmari conceived and designed the experiments, performed the experiments, authored or reviewed drafts of the article, and approved the final draft.

Ajinkya M. Pawar conceived and designed the experiments, performed the experiments, analyzed the data, prepared figures and/or tables, authored or reviewed drafts of the article, and approved the final draft.

Data Availability

The following information was supplied regarding data availability:

Raw data is available in the Supplemental Files.

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
