# Peer review of "Management of calcified canals during root canal treatment. A systematic review of case reports"

_PeerJ, doi:10.7717/peerj.19900_

## Round 0.1 · original submission · Major Revisions

Reviewer 1 ·

Basic reporting

It is generally acceptable, but major issues are listed below:

The term "BMP" is used (line 30) but not defined — presumably refers to "biomechanical preparation".
The manuscript describes the study as a “meta-analysis” (line 113), which is incorrect for a synthesis of case reports with no pooled data or statistical aggregation.
Numerous redundancies and grammatical errors detract from readability.
Phrases like “highly effective” and “significantly enhances success” are used without quantitative support.

Experimental design

The manuscript describes the study as a “meta-analysis” (line 113), which is incorrect for a synthesis of case reports with no pooled data.
There is no standard definition of calcified canals. Inclusion of cases is based on narrative descriptions in case reports without defined radiographic or clinical thresholds.
Some cases used CBCT, others only 2D IOPA radiographs, contributing to variability in diagnostic confirmation.
Several included studies lacked adequate reporting on etiology and clinical context (e.g., trauma, aging).

Validity of the findings

Most case reports were appraised as “low quality” due to poor documentation and lack of adverse event reporting.
The manuscript draws generalised clinical recommendations and success rates based on low-level evidence. There is no attempt to compare techniques or contextualise outcomes with broader clinical studies.
Despite describing itself as a meta-analysis, there is no quantitative pooling of data or effect estimation.
Descriptions of 3D guides and navigation systems are mixed without technical clarity.
Wide variation in patient demographics, canal locations, techniques used, and follow-up durations undermines the coherence of synthesis.

Reviewer 2 ·

Basic reporting

a. Language: There are few grammatical errors that occasionally make the text hard to follow. A thorough proofreading is recommended (highlighted text in PDF), (lines 85-85, lines 260-264)
b. Intro & background show clear context
c. Can add citations for some of the sentences/statements in the introduction, as highlighted in the PDF
d. Few references are not complete. The journal name is not mentioned.

Experimental design

a. Original primary research is within the scope of the journal
b. The research question is well defined, relevant & meaningful.
c. A note on the operator experience and its role in management can be included.

Validity of the findings

No comment

Additional comments

a. The CBCT was not advised in all the case reports? Could this incorporate bias in treatment planning? (Lines 224, 234)
b. Few references are not complete. The journal name is not mentioned.
c. The conclusions are well stated; however, not all points mentioned in the aims and objectives mentioned in the introductions have been addressed.

Annotated reviews are not available for download in order to protect the identity of reviewers who chose to remain anonymous.

---

## Round 0.2 · accepted · Accept

The manuscript presents a well-designed and scientifically sound study with clear methodology and relevant results. All reviewers’ concerns have been adequately addressed.

Reviewer 2 ·

Basic reporting

I am satisfied with the modifications.

Experimental design

-

Validity of the findings

-